# Brain Metabolic Connectivity Patterns in Patients with Prolonged Disorder of Consciousness after Hypoxic-Ischemic Injury: A Preliminary Study

**DOI:** 10.3390/brainsci12070892

**Published:** 2022-07-07

**Authors:** Zhijie He, Rongrong Lu, Yihui Guan, Yi Wu, Jingjie Ge, Gang Liu, Ying Chen, Hongyu Xie, Junfa Wu, Jie Jia

**Affiliations:** 1Department of Rehabilitation Medicine, Huashan Hospital, Fudan University, Shanghai 200040, China; hezhijie@fudan.edu.cn (Z.H.); 0356213@fudan.edu.cn (R.L.); wuyi4000@163.com (Y.W.); 16111220062@fudan.edu.cn (G.L.); flyriki@126.com (Y.C.); huangdaitian@hotmail.com (H.X.); 2PET Center, Huashan Hospital, Fudan University, Shanghai 200235, China; guanyihui@hotmail.com (Y.G.); lovejingjie@fudan.edu.cn (J.G.); 3National Clinical Research Center for Aging and Medicine, Huashan Hospital, Fudan University, Shanghai 200040, China; 4National Center for Neurological Disorders, Shanghai 200040, China

**Keywords:** hypoxic ischemic encephalopathy, disorders of consciousness, metabolic connectivity, positron emission tomography, ^18^F-fluorodeoxyglucose

## Abstract

Understanding the patterns of brain glucose metabolism and connectivity in hypoxic-ischemic encephalopathy (HIE) patients with prolonged disorders of consciousness (DOC) may be of positive significance to the accurate assessment of consciousness and the optimization of neuromodulation strategy. We retrospectively analyzed the brain glucose metabolism pattern and its correlation with clinical Coma Recovery Scale-Revised (CRS-R) score in six HIE patients with prolonged DOC who had undergone ^18^F-deoxyglucose brain positron emission tomography scanning (FDG-PET). We also compared the differences in global metabolic connectivity patterns and the characteristics of several brain networks between HIE patients and healthy controls (HC). The metabolism of multiple brain regions decreased significantly in HIE patients, and the degree of local metabolic preservation was correlated with CRS-R score. The internal metabolic connectivity of occipital lobe and limbic system in HIE patients decreased, and their metabolic connectivity with frontal lobe, parietal lobe and temporal lobe also decreased. The metabolic connectivity patterns of default mode network, dorsal attention network, salience network, executive control network and subcortex network of HIE also changed compared with HC. The present study suggested that pattern of cerebral glucose metabolism and network connectivity of HIE patients with prolonged DOC were significantly different from those of healthy people.

## 1. Introduction

Various causes, such as cardiac arrest and asphyxia, lead to brain injury caused by global cerebral ischemia and hypoxia in adults, which can lead to continuous disorders of consciousness (DOC). Behavioral assessment is the first-line assessment to evaluate the degree of DOC in clinical practice. At present, the most commonly used clinical scale with high sensitivity is the Coma Recovery Scale-Revised (CRS-R) [1], which is possible to distinguish between vegetative state/unresponsive arousal syndrome (VS/UWS; presence of eye opening and reflex behavior), minimally conscious state (MCS; showing repeatable but inconsistent conscious behavior, such as command following or visual tracking) and emerging from MCS (EMCS; patients recover the ability to use objects in a functional manner).

However, neurological deficits caused by hypoxic-ischemic encephalopathy (HIE), such as limb paralysis, spasm and blindness, may affect the results of bedside behavioral evaluation of DOC. In clinical practice, we found that even in VS patients, some still retained emotional response adapted to the external environment, which is not included in the evaluation of CRS-R. Through neuroimaging studies based on specific tasks, Owen AM et al. [2] and Monti MM et al. [3] suggested that even patients with VS may still have a certain degree of cortical activity. Schiff ND et al. [4] defined these DOC patients who were unable to respond to external instructions but still had reactive cortical activity as experiencing “cognitive motor dissociation” (CMD). Thibaut A et al. [5] introduced the concept of “non-behavioral MCS” (MCS star) and diagnosed DOC patients with behavioral evaluation as VS but with brain glucose metabolism preservation, indicated by Positron emission tomography (PET) imaging as MCS star. Compared with VS patients with poor brain glucose metabolism preservation indicated by PET imaging, MCS star patients may have better prognosis.

A number of neuroimaging and electrophysiological studies have shown that the structural and functional connections between bilateral frontal parietal lobe and cingulate cortex in patients with DOC have decreased [6,7], and these brain regions are closely related to arousal and consciousness [8]. The default mode network (DMN) is mainly composed of medial prefrontal cortex, anterior cingulate gyrus, posterior cingulate gyrus, precuneus and bilateral inferior parietal lobules. The functional activity of DMN in patients with DOC is significantly reduced [9]. fMRI studies suggested that the low frequency fluctuations and regional voxel homogeneity of various brain regions in DMN of patients with DOC were reduced [10], while the functional connectivity of DMN is associated with the level of consciousness [11,12,13]. The connection between medial prefrontal lobe and posterior cingulate gyrus may be related to the prognosis of DOC [14]. In addition, the functional disconnection between cortex and subcortical is also considered to be one of the biological mechanisms of DOC [15]. Patients with DOC have interruption and disorder of the effective connectivity between the left striatum and bilateral precuneus, cingulate gyrus and left middle frontal gyrus [16]. The thalamus is also an important subcortical nucleus for consciousness maintenance. The functional connectivity between medial thalamic nucleus, prefrontal lobe and anterior cingulate gyrus in VS patients decreased significantly [17].

In recent years, various brain stimulation technologies have been explored and applied to enhance the recovery of DOC patients. For example, transcranial pulsed-current stimulation of the prefrontal lobe may enhance local brain functional connection [18]; the treatment of invasive low-intensity focused ultrasound (LIFU) on the thalamus could improve the behavior of patients with DOC, and a functional magnetic resonance imaging (fMRI) study suggested that LIFU can improve the functional connection of the thalamus [19]; meta-analysis showed that deep brain stimulation (DBS) treatment may improve the CRS-R score of DOC patients in some studies, but there was diversity in the targets and parameters of DBS [20]. Therefore, understanding the brain network model of patients with persistent disorders of consciousness after cerebral ischemia and hypoxia may be of positive significance for selecting more appropriate and accurate stimulation targets and neural regulation strategies in the future.

PET imaging is safe, comfortable, and atraumatic and is currently the only imaging technique that can quantitatively evaluate biochemical alterations in vivo, accurately reflecting the biology of the lesion. The present study retrospectively analyzed six DOC patients with hypoxic-ischemic brain injury who underwent brain ^18^F-deoxyglucose PET scanning in our center, and analyzed the brain glucose metabolism pattern of these HIE patients as well as their correlation with CRS-R score. We also compared the differences in brain metabolic connectivity patterns between HIE patients and healthy subjects, as well as the metabolic connectivity characteristics of major brain networks in HIE patients.

## 2. Materials and Methods

### 2.1. Participants

Six patients hospitalized in the Department of Rehabilitation Medicine of Huashan Hospital, affiliated to Fudan University, Shanghai, China, and diagnosed with hypoxic-ischemic encephalopathy (HIE) were included in this study (male/female, 3/3; age, 45.83 ± 16.23 years). The causes of cerebral hypoxia ischemia in these HIE patients were cardiac arrest in 2 cases, anesthesia accident in abdominal surgery in 2 cases, massive bleeding of gastric cancer in 1 case (no evidence of tumor intracranial metastasis) and asphyxia due to food choking in 1 case. All of them had prolonged disorder of consciousness (DOC) after rescue. The average duration of DOC was 20.83 ± 8.93 weeks at the time of PET scanning. The state of consciousness of these HIE patients was assessed according to the Coma Recovery Scale-Revised Version (Chinese version) (CRS-R) [21,22], which was completed by two experienced neurorehabilitation experts according to the standard process of CRS-R [22]. We adopted the highest CRS-R score within 1 week before and after the time of PET scan. Among them, 5 cases were VS and 1 case was MCS. The clinical information and CRS-R assessments of HIE patients are listed in Table 1. In addition, we included 18 gender and age matched healthy people as the healthy control (HC) group. The demographic information is listed in Table 2. All subjects are right-handed. The study was approved by the Ethics Committee of Huashan Hospital and was conducted in accordance with the tenets of the Declaration of Helsinki.

### 2.2. FDG PET Scanning

All subjects were asked to fast for at least 6 h, but had free access to water before PET imaging. PET scans were performed with a United Imaging uMI510 PET/CT (United Imaging, Shanghai, China) in three-dimensional (3D) mode. A CT transmission scan was first performed for attenuation correction. The emission scan was acquired between 60–70 min after intravenous injection of 185 MBq of ^18^F-fluorodeoxyglucose. The imaging data was digitally registered onto a computer disk to create the sectional images of a patient’s brain [23]. As no arterial blood sampling was taken in this clinical imaging protocol, we used radioactivity images to measure changes in relative regional glucose metabolism. All studies in patients and normal individuals were performed in a resting state in a quiet and dimly lit room. All patients and normal individuals were monitored via cameras during the course of uptake and scanning procedure.

### 2.3. Image Preprocessing

Imaging data were processed by using Statistical Parametric Mapping (SPM8) software (Wellcome Department of Imaging Neuroscience, Institute of Neurology, London, UK) implemented in Matlab 8.3.0 (Mathworks Inc., Sherborn, MA, USA). Scans from each subject were spatially normalized into Montreal Neurological Institute (MNI) brain space with linear and nonlinear 3D transformations. The normalized PET images were then smoothened by a Gaussian filter of 8 mm FWHM over a 3D space to increase signal to noise ratio for statistical analysis.

Due to the decreased glucose metabolism in the whole brain of HIE patients, im-proper use of global signal normalization may thus lead to incorrect increase of the relative metabolic value [24]. The brainstem function of VS patients is relatively reserved, which may be related to maintaining the awakening and autonomic nerve function of VS patients [25]. Considering that midbrain atrophy in patients with DOC may be related to dopaminergic nerve dysfunction [26] and the volume of medulla oblongata is relatively small, we chose pons as a reference for radioactive count correction in each brain region.

PET imaging data were analyzed by using SPM8 software as described previously [27]. To characterize metabolic activity in HIE patients compared with controls, we performed a group comparison by using a two-sample *t*-test according to the general linear model at each voxel. The contrasts for the decreased and increased metabolism were set as (1 −1) and (−1 1). Mean signal differences over the whole brain were adjusted by radioactivity counts of pons in each individual subject. To evaluate significant differences, we set the peak threshold at *p* < 0.001 (uncorrected) over whole brain regions with an extent threshold of 100 voxels (corresponding to a tissue volume of 800 mm^3^). For a stricter criterion, we also highlighted clusters that survived a family wise error (FWE) correction at *p* < 0.05. Significant regions were localized by Talairach Daemon software (Research Imaging Center, University of Texas Health Science Center, San Antonio, TX, USA). The SPM maps for increased or decreased metabolism were overlaid on a standard T1-weighted MRI brain template in stereotaxic space.

To quantify metabolic changes in specific regions, we used a 3-mm radius spherical volume of interest (VOI) within the image space, centered at the peak voxel of clusters that were significant in the two-sample *t*-test. We then calculated the relative cerebral glucose metabolic values (i.e., adjusted by pons) in all patients and normal individuals.

We also used a multiple regression analysis to determine the relationship between clinical measures (i.e., CRS-R) and cerebral metabolic values in patients with HIE and the peak threshold was set at *p* < 0.01 (uncorrected). The contrasts for the positive and negative correlation were set as (0 1) and (0 −1). For a stricter criterion, we also highlighted clusters that survived at *p* < 0.001(uncorrected). The overlaid SPM maps and relative metabolic values were obtained using the same methods as mentioned before.

### 2.4. Metabolic Connectivity Matrix

To gain a deeper insight into potential mechanisms underlying altered glucose metabolism, we assessed region-of-interest (ROI)-based metabolic connectivity. We used the pons as reference region to count regional relative glucose metabolic values and further generated the standardized uptake value ratio (SUVR) map. The regional relative glucose metabolic activity was obtained from the 116 brain regions using the Automated anatomical labelling atlas (AAL), which were assigned to the following 8 anatomical compartments: frontal cortex, parietal cortex, temporal cortex, occipital cortex, limbic system, basal ganglia, cerebellum hemisphere, and vermis of cerebellum. Then we calculated the Pearson’s correlation between the ROIs of each group, creating a pairwise metabolic connectivity matrix (116 × 116 ROIs).

In order to further analyze the characteristics of the major brain networks of HIE patients, we selected the brain regions in AAL template related to the default mode network (DMN), salience network (SN), dorsal attention network (DAN), executive control network (ECN) and subcortex network described in previous studies [28,29,30,31,32,33] as ROIs. Then we calculated the Pearson’s correlation coefficient between these ROIs and generated a metabolic connectivity matrix for each brain network.

### 2.5. Statistical Analysis

Shapiro-Wilk test was used for normality test. Demographic data were analyzed by two-samples *t*-test or chi square test. Differences in regional metabolic values between HIE and HC groups were assessed by using two-sample *t*-tests. In addition, correlations between regional relative metabolic values and CRS-R score in HIE patients were assessed by computing Pearson’s correlation coefficients. The above statistical analyses were performed using the SPSS software (SPSS, Chicago, IL, USA) and considered significant at *p* < 0.05.

## 3. Results

### 3.1. Characteristics of Relative Brain Activity in Patients with HIE

In comparison with HC group, patients with HIE showed decreased metabolism bilaterally in the medial frontal gyrus, posterior cingulate gyrus (extending to parietal-occipital lobe), postcentral gyrus and superior temporal gyrus, and in the right anterior cingulate gyrus and right caudate nucleus, associated with increased metabolism in the bilateral anterior cerebellar lobe and left middle frontal gyrus and right lentiform nucleus (Figure 1A and Table 3).

Figure 1B shows the differences in relative metabolism values between the HIE and HC groups in the major functional regions (see Figure 1A). The decreased metabolic values in the spherical volume of interests (VOI) centered at the bilateral medial frontal gyrus, posterior cingulate gyrus, postcentral gyrus and superior temporal gyrus, and right anterior cingulate gyrus and right caudate nucleus displayed significant group differences between HIE and healthy individuals (*p* < 0.01, except for the right anterior cingulate gyrus where *p* < 0.05). The increased metabolic value in the VOIs centered at the bilateral anterior cerebellar lobe and left middle frontal gyrus also showed significant group discrimination between patients and controls (*p* < 0.01). There was a trend toward significance in the metabolic value in the VOIs centered at the right lentiform nuclei in the HIE group compared with the HC group (*p* = 0.106).

### 3.2. Correlation of Consciousness Level with Relative Brain Activity in Patients with HIE

The regression analysis yielded a positive correlation of CRS-R score in HIE patients with metabolic activity bilaterally in the medial frontal gyrus, middle frontal gyrus, anterior cingulate, and caudate, and in the left superior frontal gyrus, left inferior frontal gyrus and right tuber of cerebellum. In contrast, CRS-R correlated negatively with metabolic activity in the left lingual gyrus. (Figure 2A and Table 4). The sample plots for the six major regions from the regression analysis (see Figure 2A) are given in Figure 2B–G. The CRS-R score correlated significantly with relative metabolic values in the right medial frontal gyrus (r = 0.969, *p* = 0.002), left superior frontal gyrus (r = 0.875, *p* = 0.023), right middle frontal gyrus (r = 0.862, *p* = 0.027), left caudate (r = 0.857, *p* = 0.029), right caudate (r = 0.975, *p* < 0.001), and right tuber of cerebellum (r = 0.977, *p* < 0.001).

### 3.3. Metabolic Connectivity

The metabolic connectivity matrixes of HIE and HC groups are shown in Figure 3. Compared with the HC group, the global metabolic connectivity decreased in the HIE group. Within each anatomical partition, the internal metabolic connections of frontal, parietal, temporal, and basal ganglia decreased in the HIE group compared with the HC group, but with some degree of preservation remained. In the HIE group, the internal metabolic connections were weakened in the occipital lobe and limbic system, whereas they were enhanced in the cerebellum hemispheres and cerebellum vermis. The metabolic connectivity of the occipital lobe, limbic system, and basal ganglia with other brain regions was reduced in the HIE group compared with the HC group, whereas the cerebellum showed enhanced metabolic connectivity with the frontal, temporal, and occipital lobes.

### 3.4. Brain Network Metabolic Connectivity Patterns

Figure 4A–E demonstrate the metabolic connectivity among the brain regions related to the default mode network (DMN), salience network (SN), dorsal attention network (DAN), executive control network (ECN) and subcortex network in AAL template. In the HIE group, the metabolic connectivity of the bilateral posterior cingulate gyrus with other nodes within the DMN decreased, as did the connectivity of the bilateral precuneus and left angular gyrus with other nodes within the network (Figure 4A). In the SN, the metabolic connectivity of the bilateral insula and left anterior cingulate with other nodes within the network decreased in the HIE group (Figure 4B). In the DAN, the metabolic connectivity between brain regions within both hemispheres were preserved in the HIE group, but the interhemispheric metabolic connections between bilateral superior parietal gyrus, inferior parietal gyrus, supramarginal gyrus, and angular gyrus decreased in the HIE group compared with the HC group (Figure 4C). Similarly, in the ECN, the cross hemispheric metabolic connectivity between the nodes of bilateral parietal lobe in HIE patients decreased, and the metabolic connectivity between the nodes of right frontal lobe and left parietal lobe decreased, but the metabolic connectivity between the nodes of left frontal lobe and right parietal lobe was preserved in the HIE group. Figure 4D). In the subcortex network, the metabolic correlations between the caudate nucleus, putamen, and globus pallidus were enhanced in the HIE group compared with the HC group (Figure 4E). However, there was decreased metabolic correlation between the bilateral hippocampus and amygdala and decreased metabolic connectivity with the contralateral caudate nucleus, putamen, and globus pallidus, respectively, in the HIE group (Figure 4E). Metabolic connectivity between the bilateral thalamus and caudate nucleus was also decreased in HIE patients (Figure 4E).

## 4. Discussion

In the present study, we retrospectively analyzed the FDG-PET brain images and clinical information of six patients with hypoxic-ischemic encephalopathy with persistent DOC, and compared and analyzed the characteristics of brain metabolism in HIE patients with healthy people and the correlation between brain metabolism and the evaluation of clinical consciousness level; We also constructed the metabolic connection matrix between HIE patients and healthy people, and compared the brain metabolic connection patterns and characteristics between HIE patients with DOC and healthy people; finally, we analyzed several currently recognized brain networks of HIE patients, and discussed the connectivity mode of key nodes in the brain network of DOC patients.

Our observation of decreased metabolism in the frontal lobe and cingulate gyrus in HIE patients with the presence of DOC and the correlation between the degree of local metabolic preservation and CRS-R scores are similar to the results of previous studies on brain metabolism pattern in DOC patients [34,35,36,37]. Through the whole brain functional connectivity analysis by anatomical partition, we found that the internal metabolic connectivity in the frontal, parietal, and temporal lobes of HIE patients, although decreased compared with healthy individuals, were still preserved to some extent compared with other brain regions. However, the more pronounced decrease in metabolic connectivity of the frontal, parietal, and temporal lobes with the occipital and limbic systems in HIE suggests that cognitive processes in these patients may not be in interface with visual input and there might be a dyscoordination of sensory input to motor output from the limbs, but the processing ability of auditory input might be relatively preserved in this subset of patients. This may correspond to the “cognitive motor dissociation” (CMD) [4] or “higher-order cortex motor dissociation” (HMD) [38], as described in previous studies. In clinical practice, we also found that some VS patients had affective reactions that were compatible with the surrounding context, but this partial functional preservation in DOC patients could not be reflected in CRS-R scores. Visual and motor function partial rating items, which corresponded with the behavioral performance of MCS in the CRS-R score, both related to visual perceptual abilities. Therefore, the CRS-R score may not fully describe the level of consciousness in this subset of DOC patients with partially preserved prefrontal cortical function but with a loss of liaison between the frontal and occipital lobes. Clinical evaluation of such patients may also need to incorporate task-related imaging or electrophysiological tests such as fMRI, PET and EEG, etc., and further studies were needed to investigate the prognostic differences between such patients and VS patients without affective reactions.

In terms of several major brain networks of the cortex, we observed a significant loss of connectivity of the bilateral posterior cingulate and precuneus with bilateral medial prefrontal lobes in the DMN of HIE group, which is similar to previous findings of functional connectivity [39,40,41,42] and suggests an important role of the precuneus and posterior cingulate in the maintenance of consciousness. We also observed that there were metabolic negative correlations between posterior cingulate and medial prefrontal, suggesting a potential mutual inhibitory interaction between them. We were also able to see a trans-hemispheric loss of contact in the angular gyrus and precuneus, and the same phenomenon with the DAN, especially at various nodes located in the parietal lobe, suggesting a decreased contact between the bilateral parietal lobes in HIE group, which may be related to the decreased higher-order sensory integration in response to environmental stimuli in DOC patients [43]. The connectivity between the frontal and parietal lobes is also closely related to the maintenance of the level of consciousness [44]. In the ECN, we found that the metabolic connectivity between each node of the right frontal lobe and the left parietal lobe decreased in the HIE group, whereas the metabolic connectivity between each node of the left frontal lobe and the right parietal lobe was instead preserved, and the laterality of frontoparietal functions in DOC patients needs further studies for validation and mechanistic exploration. Recent studies have suggested that non-invasive neuromodulation techniques, mainly applied to the prefrontal lobe, may improve local cortical excitability and functional connectivity [18,45], but evidence for their efficacy on the improvement of consciousness levels was insufficient. We speculated whether the optimization of stimulation site and parameters could be performed according to the abnormal metabolic connectivity pattern of frontoparietal network in DOC patients. In addition, we found decreased metabolic connectivity between bilateral insula and various nodes of the prefrontal lobe in the SN of the HIE group. Various sensory stimuli evoke excitation in the insula, an important brain region for overall perception of the forming external environment and intrinsic self-experience [46]. After propofol anesthesia in healthy people, anterior insular function was depressed, which in turn affected the dynamic conversion of the DMN to the DAN [47]. We speculated that the insula might also serve as one of the potential targets of neuromodulation in DOC patients.

Recently, an increasing number of studies have focused on the functional connectivity between subcortical structures, as well as between cortical brain regions, in DOC patients, and a series of clinical investigations with DBS of subcortical structures in DOC patients has been performed. Our results suggested that patients with HIE who present with DOC had hypometabolism in the caudate nucleus and that the degree of hypometabolism in the bilateral caudate nucleus was significantly correlated with the level of consciousness. Metabolic connectivity analysis of subcortex networks suggested enhanced connectivity between various nuclei within the bilateral striatum, while metabolic connectivity between other subcortical structures of the thalamus was decreased. The striatum of DOC patients may have reduced inhibition of the globus pallidus, which may then lead to hyperexcitability of the globus pallidus and, subsequently, increased inhibition of the thalamus [48,49]. In addition, intracranial EEG studies have revealed that the ventral striatum may be involved in the regulation of cortical information flow, which makes it an important part of the experience of conscious perception [50]. It was also believed that the integration of the parietal lobe with striatum and thalamus contributed more than the frontal cortex in the maintenance of consciousness [15], and the metabolic connectivity matrix in our results also showed decreased connectivity between basal ganglia and frontoparietal cortex in the HIE group. Nonhuman primate studies suggested that neurons in the deep layers of the thalamus and cortex were most sensitive to changes in the level of consciousness, stimulation of the thalamus restored wakefulness in anesthetized macaques [51], and DBS to different parts of the thalamus increases both the level of arousal and self-awareness [52], suggesting that the thalamus was one of the important targets for DBS therapy in patients with DOC.

Clearly, the present study still has some limitations. First, although the included patients were all DOC patients after HIE occurrence, the sample size was indeed small. Although there were significant differences in cerebral glucose metabolic and metabolic connectivity in the HIE group compared to the healthy participants, we could no longer perform further subgroup analyses to explore the cerebral metabolic patterns and correlations with clinical scores in the VS and MCS patients. Second, in terms of PET brain imaging processing, we chose the pons as a reference for normalization of whole brain metabolic values, but also lost information on pons metabolism; moreover, there is currently no evidence that pons metabolism is not different in DOC patients compared with healthy individuals. It is possible that in future studies we could choose radioactivity counts at other sites for normalization, such as the soft tissue under the scalp, or perform simultaneous arterial blood collection and radioactivity count measurements with PET scan. Third, the results of the brain metabolic connectivity analysis of the population, which cannot represent the metabolic connectivity of individuals, were not statistically analyzed in intergroup comparisons. We performed the brain metabolic connectivity versus brain network analysis when adopting an anatomical structure based partitioning approach with a relatively small number of nodes. In further studies, we would continue to increase the sample size, as well as conduct long-term follow-up of cases to explore the association of cerebral metabolic patterns and prognosis, and perform subgroup analysis and comparison of VS and MCS; in terms of data correction, we would attempt to acquire subject blood samples for absolute value quantification of radioactivity counts, to reduce data correction bias; we would also apply PET combined with resting-state fMRI and EEG for multimodal measures of brain functional connectivity to compensate for the relative lack of temporal and spatial resolution.

## 5. Conclusions

Through PET imaging analysis of HIE patients with prolonged DOC, the present study found that patients with HIE had decreased metabolism in the frontal lobe and cingulate gyrus, and that the degree of local metabolic preservation may have a correlation with level of consciousness; the internal metabolic connections in the frontal, parietal, and temporal lobes of HIE patients were to some extent preserved compared to healthy individuals. However, there was a significant decrease in metabolic connectivity of the frontal, parietal, and temporal lobes with the occipital and limbic systems in HIE. There may be abnormal network connectivity patterns in HIE patients in several major brain networks. In the DMN, there was a significant loss of connectivity of the bilateral posterior cingulate and precuneus with bilateral medial prefrontal lobes in patients with HIE, and in the DAN and ECN there was a decline in connectivity between each node in the bilateral parietal lobes in patients with HIE, as well as a decline in metabolic connectivity between each node in the right frontal lobe and the left parietal lobe. Patients with HIE have reduced metabolic connectivity between the bilateral insula and various nodes of the prefrontal lobe in the SN. Patients with HIE may also present with dysconnectivity in subcortex structures, as well as cortico-subcortex dysconnectivity. HIE patients have hypometabolism in the caudate nucleus, and the degree of hypometabolism in the bilateral caudate nucleus correlates significantly with the level of consciousness score. Metabolic connectivity analysis of subcortical networks suggests enhanced functional connectivity between various nuclei within the bilateral striatum, but decreased metabolic connectivity of the thalamus with the other subcortical structures. The HIE group showed decreased connectivity between the basal ganglia and the parietal cortex.

## Figures and Tables

**Figure 1 brainsci-12-00892-f001:**
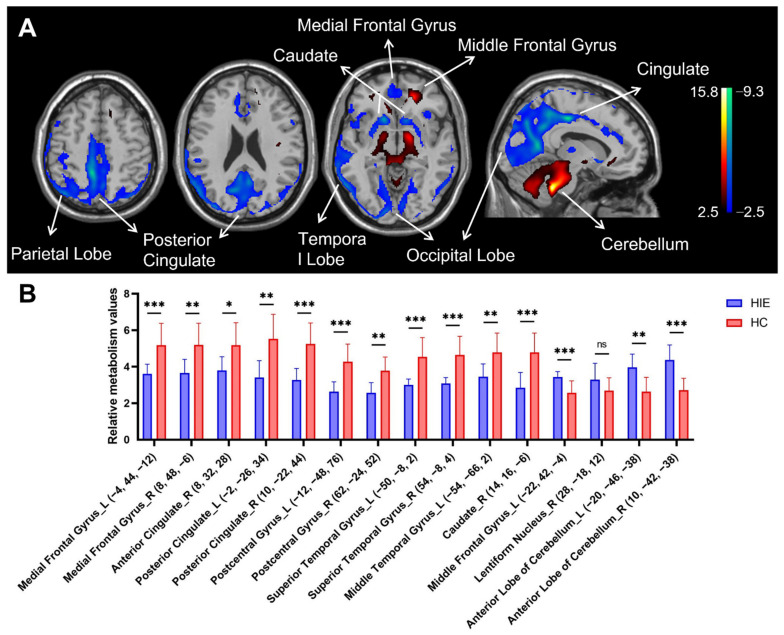
Brain regions with relative abnormal metabolism in patients with hypoxic ischemic encephalopathy (HIE) presenting with disorders of consciousness (DOC). (**A**) Normalized glucose metabolism in the HIE patients decreased (blue) bilaterally in the medial frontal gyrus, posterior cingulate gyrus (extending to parietal-occipital lobe), postcentral gyrus and superior temporal gyrus, and in the right anterior cingulate gyrus and right caudate nucleus, but increased (red) in bilateral anterior cerebellar lobe and left middle frontal gyrus and right lentiform nucleus relative to the healthy controls (HC). The overlays are depicted in neurologic orientation. The gray-scale image is a T1-weighted structural magnetic resonance imaging (MRI) that is representative of Montreal Neurological Institute (MNI) space. The thresholds of the color bars represent T values. (**B**) Differences in regional glucose metabolism in the HIE patients and HC subjects illustrated by post hoc values obtained within a spherical volume of interest (VOI) (3 mm radius) with the center coordinates of each VOI given in the parentheses. The graphs showed the relative metabolic value decreases in the bilateral medial frontal gyrus, posterior cingulate gyrus, postcentral gyrus and superior temporal gyrus, and right anterior cingulate gyrus and right caudate nucleus, while the relative metabolic value increases in bilateral anterior cerebellar lobe and left middle frontal gyrus. The error bars represent standard deviation. ns, no statistical difference. HIE, hypoxic ischemic encephalopathy. HC, healthy control. L, left. R, right. *, *p* < 0.05. **, *p* < 0.01, ***, *p* < 0.001.

**Figure 2 brainsci-12-00892-f002:**
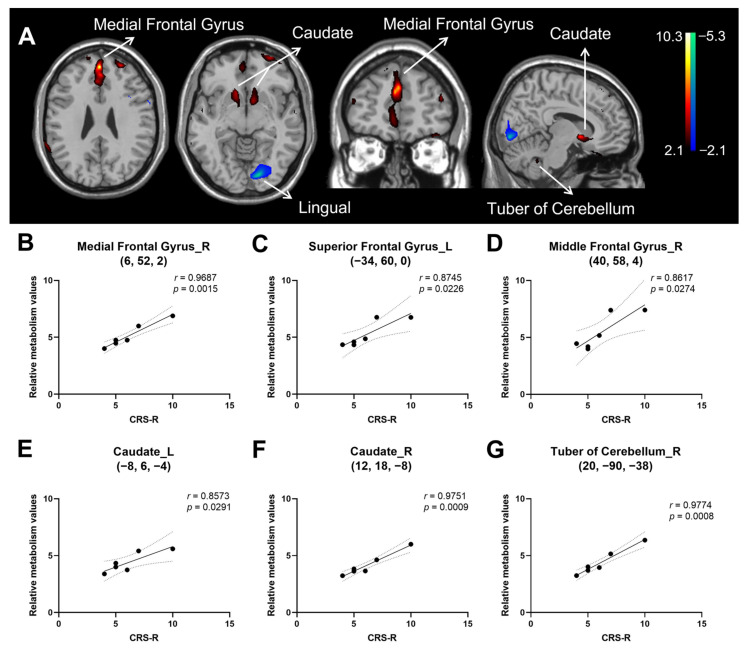
Brain regions correlated with Coma Recovery Scale-Revised (CRS-R) score in hypoxic ischemic encephalopathy (HIE) patients presenting with disorders of consciousness (DOC). (**A**) Regions with positive correlation (red) were observed bilaterally in the medial frontal gyrus, middle frontal gyrus, anterior cingulate and caudate, and in the left superior frontal gyrus, left inferior frontal gyrus and right tuber of cerebellum, whereas those with negative correlation (blue) were found in the left lingual gyrus. The overlays are depicted in neurologic orientation. The gray-scale image is a T1-weighted structural magnetic resonance imaging (MRI) that is representative of Montreal Neurological Institute (MNI) space. The thresholds of the color bars represent T values. (**B**–**G**) Correlation between relative metabolic values and CRS-R score the right medial frontal gyrus. Dotted lines represent 95% confidence bands of the best-fit line. (**B**), left superior frontal gyrus (**C**), right middle frontal gyrus (**D**), left caudate (**E**), right caudate (**F**), right tuber cerebellum (**G**). The metabolic values were obtained post hoc within a spherical volume of interest (VOI) (3 mm radius) with the center coordinates of each VOI given in the parentheses on each panel. L, left. R, right. CRS-R, coma recovery scale—revised version.

**Figure 3 brainsci-12-00892-f003:**
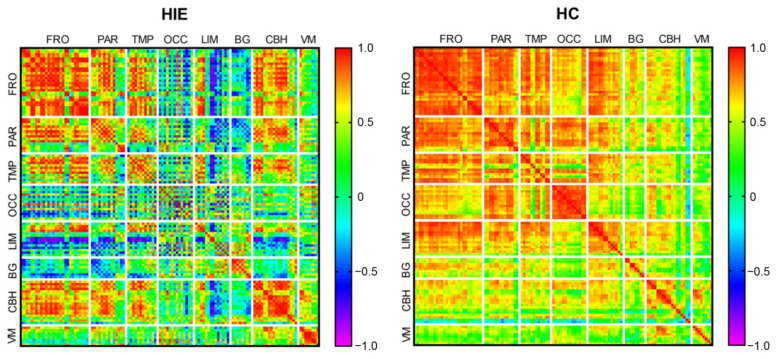
Global metabolic connectivity matrix. Colors in the metabolic connectivity matrix represent the magnitude of the correlation and colors are arranged as a rainbow color from red to purple according to the magnitude of the correlation from positive to negative. HIE, hypoxic ischemic encephalopathy. HC, healthy control. FRO, frontal cortex. PAR, parietal cortex. TMP, temporal cortex. OCC, occipital cortex. LIM, limbic structures. BG, basal ganglia. CBH, cerebellum hemisphere. VM, vermis of cerebellum.

**Figure 4 brainsci-12-00892-f004:**
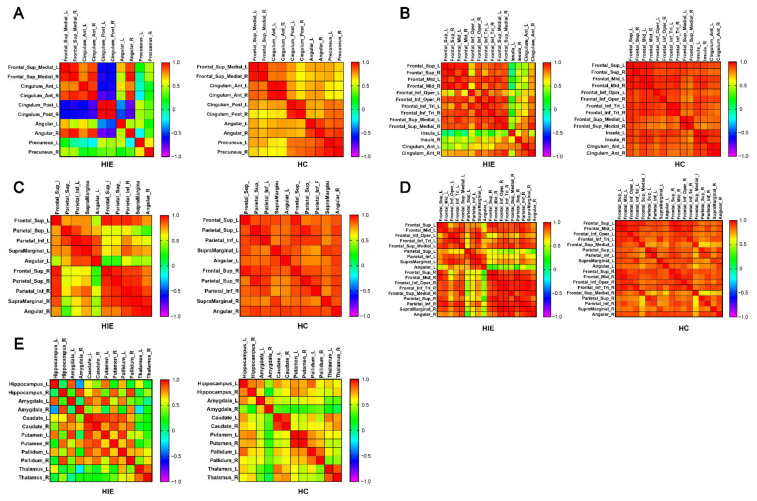
Metabolic connectivity matrix between brain regions associated with different brain networks in the AAL template. (**A**) default mode network (DMN) (**B**) salience network (SN) (**C**) dorsal attention network (DAN) (**D**) executive control network (ECN) (**E**) subcortex network. Colors in the metabolic connectivity matrix represent the magnitude of the correlation and colors are arranged as a rainbow color from red to purple according to the magnitude of the correlation from positive to negative. HIE, hypoxic ischemic encephalopathy. HC, healthy control. L, left. R, right.

**Table 1 brainsci-12-00892-t001:** Patients with HIE.

Patient No.	Gender	Age (yrs)	Time Since HIE Onset (Weeks)	Cause of HIE	Level of DOC	Total CRS-R Score
1	M	17	29	Cardiac arrest	VS	6
2	F	50	13	Anesthesia accident	VS	5
3	F	39	16	Anesthesia accident	VS	4
4	M	57	27	Massive hemorrhage ^a^	VS	7
5	M	63	32	Chocking	VS	5
6	F	49	8	Cardiac arrest	MCS	10

HIE, hypoxic-ischemic encephalopathy. DOC, disorder of consciousness. CRS-R, coma recovery scale-revised version. M, male. F, female. VS, vegetative state. MCS, minimally conscious state. ^a^ Patient 4 had a history of gastric cancer, but there was no evidence of intracranial metastasis.

**Table 2 brainsci-12-00892-t002:** Demographic data.

	HIE (*n* = 6)	HC (*n* = 18)	*p* Value
Gender (M/F)	3/3	9/9	*p* > 0.05
Age (yrs)	45.83 ± 16.23	52.67 ± 2.28	*p* > 0.05
Handedness (R/L)	6/0	18/0	*p* > 0.05

HIE, hypoxic-ischemic encephalopathy. HC, healthy control. M, male. F, female. L, left. R, right.

**Table 3 brainsci-12-00892-t003:** Brain regions with significant metabolic differences in HIE patients compared with HC subjects.

Structure	Broadmann Area	L/R	MNI Coordinates ^a^			Z_max_	Cluster Size (mm^3^)
			x	y	z		
Decreased metabolism ^b^							
Medial Frontal Gyrus	11	L	−4	44	−12	3.8	150
	10	R	8	48	−6	3.39	
Anterior Cingulate	32	R	8	32	28	3.75	122
Posterior Cingulate ^c^	23	L	−2	−26	34	5.63	10,079
(Extending to parietal-occipital lobe)	24	R	10	−22	44	5.88	
Postcentral Gyrus ^c^	5	L	−12	−48	76	5.19	118
	2	R	62	−24	52	3.84	125
Superior Temporal Gyrus	22	L	−50	−8	2	4.31	401
(Extending to insula)	22	R	54	−8	4	4.22	630
Middle Temporal Gyrus	37	L	−54	−66	2	4.09	832
Caudate ^c^	Caudate Head	R	14	16	−6	4.88	284
Increased metabolism ^b^							
Middle Frontal Gyrus ^c^	11	L	−22	42	−4	6.08	1000
Lentiform Nucleus	Putamen	R	28	−18	12	4.03	181
Anterior Lobe ^c^	Cerebellum	L	−20	−46	−38	6.21	8305
	Cerebellum	R	10	−42	−38	7.36	

MNI, Montreal Neurological Institute. FEW, family-wise error corrected, ^a^, MNI standard space. ^b^, Survived at uncorrected *p* < 0.001. ^c^, Survived at FWE *p* < 0.05.

**Table 4 brainsci-12-00892-t004:** Brain regions correlated with total CRS-R score in patient with HIE.

Structure	Broadmann Area	L/R	MNI Coordinates ^a^			Z_max_	Cluster Size (mm^3^)
			x	y	z		
Positive ^b^							
Medial Frontal Gyrus ^c^	9	L	0	50	26	3.48	104
Anterior Cingulate	32	L	0	40	26	2.79	
Medial Frontal Gyrus	9	R	6	52	2	2.43	5
Anterior Cingulate	32	R	4	32	26	2.44	3
Superior Frontal Gyrus	10	L	−34	60	0	2.84	30
Middle Frontal Gyrus	10	L	−26	52	26	2.7	12
Middle Frontal Gyrus	10	R	40	58	4	2.47	6
Inferior Frontal Gyrus ^c^	47	L	−30	32	−20	3.39	82
Caudate	Caudate Head	L	−8	6	−4	2.93	22
Caudate	Caudate Head	R	12	18	−8	2.45	11
Tuber	Cerebellum	R	20	−90	−38	2.83	42
Negative ^b^							
Lingual Gyrus	17	L	−14	−86	−6	2.75	113

MNI, Montreal Neurological Institute. L, left. R, right. ^a^, MNI standard space. ^b^, Survived at uncorrected *p* < 0.01. ^c^, Survived at uncorrected *p* < 0.001.

## Data Availability

Not applicable.

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
