# Peer review of "Brain Metabolic Connectivity Patterns in Patients with Prolonged Disorder of Consciousness after Hypoxic-Ischemic Injury: A Preliminary Study"

_brainsci, 2022, doi:10.3390/brainsci12070892_

Round 1

Reviewer 1 Report

Review Report

·      In this paper, the authors have tried to analyze the brain glucose metabolism pattern of these HIE patients as well as their correlation with CRS-R score.

·      Based on a retrospective analysis of 6 DOC patients with hypoxic-ischemic brain injury who underwent brain 18F-deoxyglucose PET scanning, the authors concluded suggested that pattern of cerebral glucose metabolism and network connectivity of HIE patients with prolonged DOC were significantly different from those of healthy people.

·      The paper is interesting, well structured, and correctly organized. The authors have clearly worked hard to detail their study, but I have minor comments:

POINTS OF STRENGTH

1. Interesting topic.

2. The results are ok.

POINTS OF WEAKNESS

1.      Retrospective analysis.

2.      Small sample size.

SPECIFIC COMMENTS

1.      What was the power of sample size calculation?

2.      What is the number of physicians who read PET images and what are their experiences?

3.      Is the reading process blind?

4.      Why you did not make interreader agreement to strengthen your results?

Reviewer 2 Report

This is a single-center, retrospective collection of patients with severe brain injury after hypoxic injury who were studied with FDG-PET (n=6). A control group provided comparisons. The pons served as the reference region, rather than evaluated with blood concentration. Comparisons were voxel-by-voxel to provide z score differences. Clustering procedures were used to provide more homogenous tagging of differences.  Clinical measures were compared to mapping, and mapping was also evaluated in terms of specified anatomical networks. 

I have two general concerns: 

1)    The mapping displayed in Figure 1, given the low number of hypoxic patients, appears scattered and noise-like, and it seems that certain areas labelled as hypometabolic, especially posterior to the central sulcus, appear nearly homogenous as atrophy may appear. In other words, with this low number of subjects, I am not convinced that hypometabolism mapped by FDG-PET confers any particular specificity other than gross cortical atrophy. 

2)    Given this scattered, possibly random labelling in the setting of a low number of subjects, the network hypotheses that is dependent on reproducible labelling of hypometabolic regions remain suspect. 

My conclusion is that a great deal of careful and thoughtful work has concluded that, essentially, patients with chronic impaired consciousness who have been injured by hypoxic brain injury have regions of hypometabolism on FDG-PET.  The specificities of regions of hypometabolism and their concordant linkages do not appear significant physiologically despite their statistical relationships. 

Reviewer 3 Report

Undoubtedly, understanding the patterns of brain glucose metabolism and connectivity in hypoxic-ischemic encephalopathy patients with prolonged disorders of consciousness may be of positive significance to the accurate assessment of consciousness and the optimization of neuromodulation strategy. 

My notes for publication are as follows:

- I propose to extend the background in the field of methods of acquisition and archiving of data from the human brain by quoting, for example: Methods of acquisition, archiving and biomedical data analysis of brain functioning, Biomedical Engineering and Neuroscience, Proceedings of the 3rd International Scientific Conference on Brain-Computer Interfaces, BCI 2018, March13-14, Opole, Poland, Advances in Intelligent Systems and Computing book series (AISC, volume 720).

- The article should be expanded in the Statistics Analysis section. The current description is skimpy. Please explain the application of such and not other statistical methods to this medical data.

- In my opinion, Conclusions should be extended to include plans for the future in the field of research.

Round 2

Reviewer 3 Report

Dear Authors, 

Thank you for the changes made.

I accept the submitted explanations.

Everything is fine now. I recommend the article for publication.

Author Response

We are very grateful for your comments and suggestions.